Integration of phylogenomics and molecular modeling reveals lineage-specific diversification of toxins in scorpions

Santibáñez-López Carlos E. 1 santibanezlo@wisc.edu
http://orcid.org/0000-0002-1138-7533 Kriebel Ricardo 2
Ballesteros Jesús A. 1
Rush Nathaniel 3
Witter Zachary 3
Williams John 3
Janies Daniel A. 3
http://orcid.org/0000-0002-2328-9084 Sharma Prashant P. 1 prashant.sharma@wisc.edu
1 Department of Integrative Biology, University of Wisconsin-Madison , Madison, WI , USA
2 Department of Botany, University of Wisconsin-Madison , Madison, WI , USA
3 Department of Bioinformatics and Genomics, University of North Carolina at Charlotte , Charlotte, NC , USA
Crandall Keith
Electronic publication date: 2018 Nov 14
Publication date: 2018
Volume: 6
Electronic Location ID: e5902
Received 2018 Aug 9; Accepted 2018 Oct 9
Copyright: © 2018 Santibáñez-López et al.
Copyright year: 2018
Copyright holder: Santibáñez-López et al.
License: This is an open access article distributed under the terms of the Creative Commons Attribution License, which permits unrestricted use, distribution, reproduction and adaptation in any medium and for any purpose provided that it is properly attributed. For attribution, the original author(s), title, publication source (PeerJ) and either DOI or URL of the article must be cited.
License URL: https://creativecommons.org/licenses/by/4.0/

Keywords: 3D structure, Negative selection, Evolutionary shifts, Venom transcriptome, Morphometrics

Funding: The National Science Foundation 507 IOS-1552610 A postdoctoral CONACYT reg. 207146/454834 A postdoctoral award DEB-1655611 This material is based on work supported by the National Science Foundation under Grant 507 IOS-1552610 awarded to Prashant Sharma. Carlos Eduardo Santibáñez López was supported by a postdoctoral CONACYT grant (reg. 207146/454834). Ricardo Kriebel was supported by a postdoctoral grant (award DEB-1655611). The funders had no role in study design, data collection and analysis, decision to publish, or preparation of the manuscript.

==============================
Scorpions have evolved a variety of toxins with a plethora of biological targets, but characterizing their evolution has been limited by the lack of a comprehensive phylogenetic hypothesis of scorpion relationships grounded in modern, genome-scale datasets. Disagreements over scorpion higher-level systematics have also incurred challenges to previous interpretations of venom families as ancestral or derived. To redress these gaps, we assessed the phylogenomic relationships of scorpions using the most comprehensive taxonomic sampling to date. We surveyed genomic resources for the incidence of calcins (a type of calcium channel toxin), which were previously known only from 16 scorpion species. Here, we show that calcins are diverse, but phylogenetically restricted only to parvorder Iurida, one of the two basal branches of scorpions. The other branch of scorpions, Buthida, bear the related LKTx toxins (absent in Iurida), but lack calcins entirely. Analysis of sequences and molecular models demonstrates remarkable phylogenetic inertia within both calcins and LKTx genes. These results provide the first synapomorphies (shared derived traits) for the recently redefined clades Buthida and Iurida, constituting the only known case of such traits defined from the morphology of molecules.

Introduction

Scorpions are an iconic group of arachnids that are central to investigations of arthropod terrestrialization, morphological stasis, and diversification of body plans (Kjellesvig-Waering, 1986; Jeram, 1998; Sharma et al., 2014; Waddington, Rudkin & Dunlop, 2015). To scientists and laypersons, scorpions are particularly fascinating for the diversity of their venom, a complex mixture of bioactive compounds (e.g., peptides, proteins) secreted in specialized organs and used to disrupt biochemical and physiological processes in target organisms (King & Hardy, 2013; Casewell et al., 2013; Haney et al., 2016). Among scorpions, venoms are rich in toxins with a broad array of biological targets, including those affecting Na+, K+, Cl− and Ca2+ ion channels (Possani et al., 1999; Sunagar et al., 2013; Santibáñez-López & Possani, 2015).

The origin of toxins in animal venom has been inferred to be the result of recruitment of paralogs of ancestral housekeeping genes, followed by diversification and neofunctionalization, a process driven by positive selection (Juarez et al., 2008; Fry et al., 2009; Rokyta et al., 2011; Wong & Belov, 2012; Haney et al., 2016; Dowell et al., 2016). While novel peptides often preserve the same molecular scaffold of their ancestral protein, key changes in functional residues, mostly in surface-exposed sites, acquire newly derived biological activities (Fry et al., 2009; Casewell et al., 2013). Nevertheless, two peptides with statistically insignificant sequence similarity can also adopt the same scaffold (Orengo, Jones & Thornton, 1994), resulting in evolutionary convergence in fold structures, and thus rendering inference of homology non-trivial.

The study of scorpion venom diversity is further complicated by the patchiness of existing taxonomic sampling. The advent of current-generation sequencing technology has greatly advanced the discovery of venom diversity through the availability of transcriptomes and the first scorpion genomes. However, data acquisition strategies asymmetrically favor the taxonomic sampling of Buthidae, the largest of the 20 described scorpion families (of the approximately 2,400 described species of scorpions, 48% are buthids). Buthidae are also intensely sampled because this family contains nearly all scorpion species of medical significance. Comparatively fewer resources exist for the remaining 19 scorpion families, and these have revealed additional molecular diversity that is not reflected in Buthidae (Santibáñez-López et al., 2016, 2017).

One such example of this diversity are calcins (ryanodine receptor ligands), a group of inhibitor cystine knot (ICK)-stabilized peptides found in scorpion venom. Calcins rapidly activate ryanodine receptors (RyRs) in cardiac or skeletal muscle cells in mammals with high affinity and specificity (reviewed in Xiao et al. (2016)). These peptides bind to cell surface glycosaminoglycans and membrane lipids (Mabrouk et al., 2007; Ram et al., 2008), translocating also into cells, and undergo posttranslational modifications by target cell enzymes (Ronjat et al., 2016). To date, calcins have only been discovered in 16 species from eight families (but not Buthidae). Intriguingly, calcins share phylogenetic affinities with the lambda potassium channel toxins (hereafter “LKTx”) (Gao et al., 2013; Santibáñez-López et al., 2016) only found in species of Buthidae. LKTx inhibits the K+ channel in insects without the inactivation of the skeletal-type Ca+2 RyRs in mammals. It is therefore unknown when calcins evolved in the scorpion tree of life; limitations in sampling preclude inference of whether calcins are prone to evolutionary loss and/or replacement by other toxins in such lineages as buthids.

A separate impediment to analysis of venom evolution within a phylogenetic context is a series of recent changes in understanding of scorpion basal relationships, precipitated by the advent of phylogenomic datasets (Soleglad & Fet, 2003; Coddington et al., 2004; Sharma et al., 2015). The phylogenomic tree of Sharma et al. (2015) fundamentally changed the systematics of the two basal branches of the scorpion tree of life, suggesting that the sister groups of Buthidae were the relictual south-east Asian families Chaerilidae and Pseudochactidae (which is also found in parts of central Asia). This work was nevertheless limited to 25 species and focused on a different subset of taxa from the one wherein calcins and LKTx peptides have been reported. A recent approach using a different molecular dataset (ultraconserved elements) also recovered further differences in derived parts of the tree topology, but were limited to six scorpion species, and did not include either Chaerilidae and Pseudochactidae (Starrett et al., 2017). The present ambiguity of scorpion basal relationships thus hinders reconstruction of venom evolution.

To achieve a comprehensive, integrated understanding of scorpion phylogeny and calcin/LKTx evolution, we inferred scorpion relationships using the most comprehensive phylogenomic dataset to date, maximizing the overlap between phylogenetically distinctive lineages and existing datapoints for ICK peptides. We separately surveyed venom gland transcriptomes to discover and map the distribution of ICK homologs. To assess the evolutionary dynamics of calcins and LkTx peptides, we inferred three-dimensional molecular models and performed parametric comparative analyses of protein folding and biochemical properties. Here we show that calcins and LKTx are reciprocally restricted to the two most basally branching clades of scorpions and exhibit marked phylogenetic inertia in protein shape. The identification of LKTx in venom gland transcriptomes of Buthidae, Chaerilidae and Pseudochactidae validates the monophyly of this group and constitutes the first molecular synapomorphy uniting this parvorder, Buthida.

Materials and Methods

Taxon sampling and orthology inference

We assembled a dataset of 55 scorpion species and 13 chelicerate outgroups, consisting of one complete scorpion genome, 13 EST libraries, six 454-pyrosequencing transcriptomes, and 49 Illumina transcriptomes (two newly generated for this study and 44 previous libraries generated by our research group; Table S1). Specimens of Kolotl magnus and Urodacus elongatus were dissected into RNAlater solution (Ambion, Foster City, CA, USA). Total RNA was extracted using the Trizol Trireagent system (Ambion Life Technologies, Waltham, MA, USA). Libraries were constructed in the Apollo 324 automated system using the PrepX mRNA kit (IntegenX, Pleasanton, CA, USA), with samples marked with unique indices to enable multiplexing. Concentration of the cDNA libraries was measured using the dsDNA high sensitivity (HS) assay in a Qubit v 2 fluorometer (Invitrogen, Carlsbad, CA, USA). Library quality and size selection were checked using an Agilent 2100 Bioanalyzer (Agilent Technologies, Santa Clara, CA, USA) with the HS DNA assay. The samples were run using the Illumina HiSeq 2500 platform with paired-end reads of 100 or 150 bp at the FAS Center for Systems Biology at Harvard University. De novo assemblies were conducted using Trinity v 2.8, keeping a path reinforcement distance parameter of 75 (Grabherr et al., 2011). The search for orthologous sequences to infer species trees was also conducted de novo using a phylogenetically informed orthology criterion, as implemented in UPhO v 1.0 (Ballesteros & Hormiga, 2016). The sequences of four representative species, U. planimanus, Anuroctonus phaiodactylus, Mesobuthus martensii and Chaerilus celebensis were combined, and used as query against the database containing all species in this study using the algorithm blastp. The representative species strategy was preferred over “all versus all” searches to avoid the computational burden imposed by exhaustive pairwise sequence comparisons. Sequences were clustered in gene families using mcl (Dongen, 2000; Enright, Van Dongen & Ouzounis, 2002). A variety of values for the inflation parameter was explored i = {1.4, 2, 4, 6} and the clustering produced with i = 6 was selected from the alternative clustering based on the efficiency scores reported by mcl. A total of 9,930 clusters produced with at least 20 species was carried for downstream analyses.

Gene-family trees were estimated for each cluster using FastTree v 2 (Price, Dehal & Arkin, 2010) following multiple sequence alignment (MSA) with MAFFT v 7.0 with the parameters –anysymbol and –auto (Katoh & Standley, 2013). Masking of ambiguously aligned regions was performed using trimAl v1.2 with the –gappyout algorithm (Capella-Gutiérrez, Silla-Martínez & Gabaldón, 2009) and removing sequences that after trimming had less than 50 amino acids or less than 25% unambiguously aligned sites. Parallelization of the phylogenetic pipeline was implemented through gnu-parallel (Tange, 2011). The resulting gene family trees were analyzed in search of groups of orthologs with at least 19 different species using UPhO (orthology inference parameters: –m 19 –S 0.75 –iP; Ballesteros & Hormiga, 2016), resulting in 3,110 orthologs. The individual orthogroup alignments were concatenated in a supermatrix with geneStitcher.py (Ballesteros & Hormiga, 2016). In-paralogs, alleles, duplicates and/or splice-variants retained in the orthologs were resolved in favor of the longest sequence. The resulting matrix (henceforth, Matrix 1), consisted of 3,110 loci, 792,210 aligned amino acid sites, and 69% missing data.

Phylogenomic inference and molecular dating

Maximum likelihood (ML) analyses were performed using ExaML v 3.0 (Kozlov, Aberer & Stamatakis, 2015) and IQ-TREE v 1.5.5 (Nguyen et al., 2014), implementing ultrafast bootstrap resampling to gauge nodal support (Minh, Nguyen & Haeseler Von, 2013). The matrix was partitioned by locus, selecting the best-fitting amino acid substitution model per partition (based on the automatic model assignment obtained during the ExaML run with the default ML criterion) and with per-site rates (ExaML) or with four gamma categories (in IQ-TREE). The resulting ML tree was calibrated for downstream analyses using penalized likelihood (Sanderson, 2002) as implemented in the chronos function of the R package ape (Paradis, Claude & Strimmer, 2004; Paradis, 2013), under the relaxed and correlated models and a family of values for lambda = {0.5, 1.0}. Age constraint strategy follows the calibration point previously used and justified in our recent works (Santibáñez-López, Kriebel & Sharma, 2017; Sharma et al., 2018), setting the lower bound of the crown age of Opiliones to 411 Myr; the crown age of Araneae to 305 My; the stem age of Amblypygi to 385 Myr and the stem age of Buthidae to a minimum of 120 Ma (see Sharma et al., 2018). Alternative dating strategies (e.g., Bayesian inference (BI) with explicit clock models) were explored previously in our recent work on scorpion basal diversification, and yielded congruent results with penalized likelihood using the complete matrix (ref. Sharma et al., 2018).

To explore the trade-off between matrix size and matrix completeness, three additional matrices were constructed by varying the inclusion threshold for minimum number of species present per locus. To maximize informativeness in the concatenated analysis, we treated our original matrix (Matrix 1, above) with the Matrix Reduction algorithm, which eliminates uninformative genes and taxa, using default parameters (Meusemann et al., 2010). The resulting dataset was termed Matrix 2 and consisted of 43 terminals and 1,632 genes (30% missing data). The search of groups of orthologs was conducted using a different number of species (parameter–m in UPhO). An intermediately stringent matrix required the orthologs to be present in at least 35 species (Matrix 3; 66 terminals; 799 genes; 39% missing data). Lastly, the most stringent matrix required orthologs to be present in at least 45 species (Matrix 4; 52 terminals, 108 genes; 13% missing data).

As concatenation methods can mask phylogenetic conflict when strong gene tree incongruence is incident, we conducted species tree estimation of the constituent orthologs of the four matrices mentioned above, and an additional matrix, which required only five terminals to be present per locus (Matrix 5; 25,997 genes) using the gene trees generated with FastTree2 (as mentioned above) and ASTRAL-II (Mirarab & Warnow, 2015). While molecular dating analyses were explored for Matrix 2 (selected as a compromise between higher number of loci and lower values of missing data) using PhyloBayes-mpi v 1.5 (Lartillot et al., 2013) under a CAT + GTR + (Γ model, analyses failed to converge after over 2 months of computation time on four independent runs on eight processors; this analysis was excluded from the study.

Gene tree analysis of ICK peptides

Inhibitor cystine knot homologs were retrieved from the complete dataset used in the scorpion phylogenetic analyses, as well as from GenBank and UniProt (Table S2). The signal peptides, propeptides and mature peptides were predicted using SpiderP from Arachnoserver (Herzig et al., 2010). Outgroup taxa for gene tree analysis consisted of three spider calcium channel toxins (ICK peptides). To root the tree, two disulfide-directed β-hairpin (DDH) scorpion toxins were selected: Phi-liotoxin isolated from the venom of Liocheles waigiensis (Smith et al., 2013) and the DDH-Uro-1 deduced from a cDNA cloned from U. manicatus (Sunagar et al., 2013). The phylogenetic relationship of scorpion DDH and ICK peptides has been discussed elsewhere (Smith et al., 2013; Sunagar et al., 2013). Multiple sequence alignments for the full precursor were generated using MAFFT v 7.0 (Katoh & Standley, 2013), resulting in a matrix consisting of 64 terminals and 132 amino acid sites. BI analysis was performed with MrBayes v 3.2.2 (Ronquist et al., 2012) using the Dayhoff model selected under the Bayesian information criterion, as selected by ProtTest v 3 (Darriba et al., 2011). Four runs, each with four Markov chains were implemented for 1 × 107 generations using default priors and discarding 1 × 106 generations using default priors.

Molecular models

Multiple sequence alignments of the mature peptide nucleotide sequences were generated based on their corresponding mature peptide amino acid sequences using PAL2NAL v 14 (Suyama, Torrents & Bork, 2006). The resulting codon alignment was further used for synonymous and non-synonymous substitution rates using the following models implemented in the Datamonkey server (http://datamonkey.org; Weaver et al., 2018): (a) aBSREL (Smith et al., 2015) to test whether a proportion of branches has evolved under positive selection; (b) FUBAR (Murrell et al., 2013) to provide additional support to the detection of sites evolving under positive or negative selection; and (c) MEME (Murrell et al., 2012) to detect episodic or diversifying selection at individual sites in the amino acid sequence. Statistical outcomes under all three approaches are presented in Tables S3 and S4.

Three-dimensional structures were generated for 41 calcin mature peptide sequences using the 3D structure of imperacalcin (Lee et al., 2004) in the SWISS-MODEL server (Biasini et al., 2014). The solvent accessible surface area (SAS) for all models was generated using the Adaptive Poisson-Boltzmann Solver (APBS) method (Baker et al., 2001) using the PDB2PQR server (Dolinsky et al., 2004). All 3D images were generated and visualized using PyMol v 1.8.2.1. The Accessible Surface Ratio (ASR) was calculated with GetArea (Fraczkiewicz & Braun, 1998). The molecular weight and volume were calculated based on the pdb files generated using VADAR v 1.8 server (Willard et al., 2003). Other physical and chemical characteristics (e.g., net charge) were calculated based only on the mature peptide sequences using the online peptide calculator (http://www.chinapeptides.com/english/tool.aspx) and the ProtParam server (http://web.expasy.org/protparam/) listed in the Tables S5 and S6.

Parametric analyses of molecule shape

Static images of the frontal and lateral views (as established by Xiao et al., 2016) of the 3D models of all ICK peptides (LKTx and calcins) were generated with PyMol. Files were aligned and exported as png for consistency. Outlines of these frontal and lateral images were extracted from their original images using GIMP v 2.8 (http://www.gimp.org) and converted into monochromatic jpeg files. The geometric morphometric technique of elliptic Fourier analysis (EFA) with the R package Momocs (Bonhomme et al., 2014) was applied to calculate the morphological shape variation of the outlines. Outlines were imported into R, converting them into lists of coordinates as described previously (Santibáñez-López, Kriebel & Sharma, 2017). For all structures, 20 harmonics were enough to achieve 99% of harmonic power during the EFA. The resulting coefficients were summarized using principal components analysis, which were used to visualize variation in morphospace. These coefficients were also used to calculate mean protein shapes for comparison between groups. For these comparisons, we used the tp_iso function in Momocs, which calculates and graphs deformations between two configurations as heatmaps. Specifically, we compared the mean shape between the calcins and LKTxs, as well as the different major clades within calcins.

In order to place protein morphology in a phylogenetic context, we matched the resulting principal components to the dated molecular phylogeny of scorpions for downstream analyses. This phylogeny was culled to retain the intersection of terminals for which shape data were available at the genus level. This strategy reflects low, or lack of, variation of calcin peptide sequences at the intrageneric level (e.g., the putative calcin sequences of the three species of the genus Urodacus were 100% similar). Visualization of morphospace was conducted using inbuilt functions in Momocs and phytools (Revell, 2012).

Inference of evolutionary rates

We investigated the evolutionary dynamics of protein traits by testing for shifts in trait regimes using a continuous multivariate Ornstein–Uhlenbeck (OU) approach implemented in the R library l1ou (Khabbazian et al., 2016). For this analysis, the main axes of shape variation (PC1 and PC2 of the frontal and lateral view from the morphometric analyses) were added to three other characteristics of the proteins: net charge, molecular volume, and molecular weight, for a total of seven columns of data describing the proteins. The data were mapped to the phylogeny and l1ou was used to estimate the best shift configuration and paint the edges of the phylogeny according to their corresponding regime. To select the best shift configuration, we used the phylogenetic Bayesian Information Criterion (pBIC), which has been shown to be more conservative in assigning regime shifts than the commonly used Akaike Information Criterion. We assessed statistical support for regime shifts in l1ou with 100 bootstrap resampling replicates.

Scorpion venom database

To facilitate access to the venom gland transcriptomes that we generated, we established a newly created scorpion venom database online at http://venom.space. Source files for this database consisted of 75 inputs. For each scorpion species, an interpro data file, a .fasta file and .faa file were created. These files contained headings that were used to establish the structure of the database. After input files were parsed appropriately, they were then used to populate the SQL database and a Django framework was used to create a web application with search functionality. The database includes functionalities for BLAST searches, which users can use to query a specific sequence of interest, and for MSA.

Results

Scorpion phylogenomic tree and divergence time estimation

De novo inference of orthology (i.e., analysis of gene family trees searching for groups of orthologs, with at least 19 different species, with 75% of support), resulted in the retention of 3,110 orthologs. The 3,110 genes were concatenated into a supermatrix consisting of 792,210 amino acid sites and 69.44% missing data (“Matrix 1”). The resulting phylogenomic tree supported the reciprocal monophyly of the two basal clades Buthida and Iurida (Fig. 1; Fig. S1) with maximal nodal support and stability. Within parvorder Buthida (the families Buthidae, Chaerilidae and Pseudochactidae), a clade comprised of Chaerilidae + Pseudochactidae was recovered as the sister group of Buthidae. Divergence time estimation place the diversification of crown group scorpions between 430 and 303 Mya (Figs. S2–S5). Interestingly, the family Hemiscorpiidae was placed as nested in the non-monophyletic Scorpionidae, but with insignificant branch support (Fig. 1; Fig. S1).

Figure 1 Scorpion tree of life.

(A) Maximum likelihood tree topology recovered from the analysis of 3,110 orthogroups, with 55 scorpion species and 13 outgroups. Bars to the right of terminals indicate number of orthologs. Shaded squares in Navajo plots indicate recovery of the node in the corresponding analysis with M = Matrix followed by its number (except M1e = Matrix 1 analyzed with ExaML), and colored as follows: blue squares = IQTree; pink square = ASTRAL (see also Fig. S1). Representative scorpion species from the two parvorders: (B) juvenile of Troglokhammouanus steineri; (C) adult female Isometrus sp.; (D) adult male Iurus dekanum; (E) adult male Superstitionia donensis; (F) subadult female Liocheles australasiae; (G) adult male Opisthacanthus madagascariensis; (H) adult female Kolotl magnus; and (I) adult female Megacormus gertschi. Photographs (B–F) by Gonzalo Giribet (with permission), and (G–H) by Carlos E. Santibáñez-López.

Our analyses of additional three supermatrices with different thresholds of gene occupancy (Matrices 2,3,4, see Methods) and our species tree analyses with ASTRAL using five different compositional matrices (Matrices 1–5), all recovered similar tree topologies (Fig. S1).

Gene tree topology and molecular evolution of ICK peptides

The 59 ICK homologs were represented by 52 scorpion species in 16 families. No ICK peptide sequences were discovered in Liocheles australasiae, Paravaejovis spinigerus, Centruroides (four spp.), Tityus (three spp.), Isometroides vescus and Lychas buchari. BI analyses of a matrix of the 59 ICK homologs (132 amino acid sites) recovered a gene tree subdivided into 41 calcins and 18 LKTx peptides (Fig. 2A). Our results supported some species-specific duplications of a few calcin orthologs, as inferred from clustered pairs of non-identical calcin sequences in the venom of Superstitionia donensis, Brotheas granulatus and Opistophthalmus carinatus. By contrast, the two copies of calcin sequences of Hadogenes troglodytes, plus the two copies of LKTx peptides from Isometrus maculatus, were recovered as out-paralogs. Oddly, one of the copies of H. troglodytes (accession number A0A1B3IJ19) was 100% identical to the calcin peptide sequence obtained from Scorpiops jendeki (accession number GH548250; Fig. 2A). Comparison of nucleotide sequences of these two calcins revealed only five synonymous changes across 222 basepairs. To detect the direction of selection acting on the codon sequences of LKTxs and calcins, we used several methods for inferring selection pressure implemented in the server Datamonkey. MEME aims to detect sites evolving under positive selection at a proportion of the branches, but not the entire phylogeny, whereas FUBAR assumes the selection pressure for each site is constant across the entire phylogeny. FUBAR found evidence of episodic negative/purifying selection at 4 (LKTx) and 19 (calcin) sites with posterior probability of 0.99 (Figs. 3A and 3B; Table S3); MEME found one site under the pressure of positive selection in the LKTx sequences, but none in the calcin sequences (Figs. 3C and 3D; Table S4). This same site was detected as evolving under neutral evolution with FUBAR, but without statistical significance.

Figure 2 Calcin and LKTx evolutionary analyses.

(A) Evolutionary tree topology of the ICK peptides recovered from the Bayesian inference analysis of 59 sequences of LKTx (Buthida) and calcins (Iurida) isolated or deduced from cDNA cloned from venom of scorpion species. Consensus amino acid sequences of LKTx (B–D) and calcins (E) showing the highly conserved disulfide bridges (lines) formed by six cysteines. Cysteines in red have their ASR less than 20% exposed, cysteines in orange have their ASR more than 30% but less than 50% exposed, and cysteines in blue have their ASR more than 51% exposed. Representative three-dimensional model of a calcin projected with the SAS corrected with APBS; frontal (F) and lateral (G) surfaces. Key amino acid used as landmarks to align the images indicated by their single letter code and their position in the MSA.

Figure 3 Molecular evolutionary analyses.

(A and B) Site selection analyses of the LKTx (A) and calcin (B) sequences with FUBAR. Visualization of the difference between the values of α (red) and β (cyan). Light red indicates sites evolving under negative selection with probabilities greater than 0.90. (C and D) Site selection analyses of the LKTx (C) and calcin (D) sequences with MEME as a function of the visualization of the difference between the values of α (synonymous substitution ratio, in red) and b+ (non-synonymous substitution at a site for the positive/neutral evolution component in cyan). Asterisk indicates the site with β+ value greater than α with a p > 0.90. Other statistics are found on Tables S3 and S4.

Molecular morphology reveals phylogenetic inertia of ICK peptides

The 3D models of 41 calcin mature peptide sequences were generated using the 3D structure of imperacalcin, a calcin isolated from the venom of Pandinus imperator (Pdb file 1IE6) as a template. Additionally, to compare the morphology of calcins to that of the LKTx peptides, an additional 17 models were generated: 13 models for sequences from the family Buthidae, two from Chaerilidae, and two from Pseudochactidae. We characterized the ASR of the amino acid residues using GetArea (Fraczkiewicz & Braun, 1998). Our results showed four of the six cysteines were buried forming the core of the protein (ASR less than 20% exposed; Figs. S6 and S7), suggesting the disulfide bridges were not altered by the insertion/deletion of amino acids between cysteines (Fig. 2B). Up to seven residues formed the core of the protein model of the LKTx peptides (Fig. S6). In contrast, five residues (four cysteines) were buried forming the core of calcins (Fig. S7), although some calcins had up to seven residues buried (the two species of Bothriurus).

To elucidate the evolution of these peptides, 3D models with SAS (Fig. 2C; Figs. S6 and S7) for two static image views (frontal and lateral) were generated using the APBS method (Baker et al., 2001) and the PDB2PQR server (Dolinsky et al., 2004). Morphometric analysis of molecular models showed that calcin shape is distinct from the LKTx peptides across multiple principal components (Fig. 4). In the EFA of the frontal SAS of calcins and LKTx peptides, 49% of the variation is explained by PC1 (p < 0.01), and showed that calcins have a well-defined apex (corresponding to extra amino acids before the first cysteine) and are consistently slenderer than the LKTx, which are comparatively globular and lack the distinctive tip. PC2 explained 13% of the variation (p = 0.17) and showed the differences between the length of the left-anterior margin of the calcins, and the right-anterior margin of the LKTx. For the lateral view, EFA of the lateral SAS of calcins and LKTx, PC1 explained 67.5% of the variation (p < 0.01), and showed that calcins have two distinctive apexes (top and bottom), whereas LKTx lack them. A comparatively globular shape is present on the LKTx peptides in contrast with a slenderer shape of calcins. PC2 explained less than 10% of the variation (p = 0.42) and segregated the shape of three calcins based on the presence of a slender apex in their model projection, as a result of a longer amino acid sequence (35, including more amino acids before the first cysteine).

Figure 4 Morphometric analyses of the 3D structure of ICK peptides in scorpion venom.

(A and B) Visualization of phylogenomic tree on the morphospace of the frontal (A) and lateral (B) shape data, showing the distinction between LTKx peptides (orange) and calcins (purple); horizontal axis indicates PC1 values and vertical axis indicates PC2 values. (C and D) Visualization of PC1 values of the frontal (C) and lateral (D) shape data as a function of phylogenetic relationships recovered from the dated molecular tree; horizontal axis indicates the time of divergence and vertical axis indicates the PC1 values.

Superimposition of the phylogram onto the principal components of scorpion ICK peptides showed that these homologs reflect the partitioning of molecular morphospace with high fidelity (Figs. 4A and 4B), and that this partitioning mirrors the basal split between Buthida and Iurida. While calcins were segregated by PC1 of the frontal view of their 3D structure, LKTx peptides showed great shape plasticity across PC1, but were more restricted in PC2. Furthermore, PC1 of the lateral view also distinguishes calcins from LKTx peptides, but PC2 did not help separate these two groups. Phenograms of PC1 for both views also suggest that the distinction between calcins and LKTx has persisted for a long span of evolutionary time (Figs. 4C and 4D).

Furthermore, the differences we observed in molecular shape between calcins and LKTx were not simply a manifestation of the size of the molecule. We examined two other biochemical properties of peptides, molecular weight and molecular volume. Phenograms of these properties showed broad overlap in the molecular weight and volumes of calcins and LKTx peptides (Figs. S8 and S9). Similarly, we found no significant differences in the molecular weight (t = −0.71; p = 0.49) and molecular volume (t = −0.84; p = 0.41) of calcins and LKTx peptides, although we note that calcin molecular weight is less variable than the molecular volume. These results suggest that the more globular shape of LKTx does not result from a simple difference in the size of calcins and LKTx.

Further analyses support the inference that principal components of molecular shape are able to reflect functional properties. We tested the correlation between PC1 of both surfaces and a third biochemical property, net charge (Fig. 5). We discovered that net charge differs significantly between calcins and LKTx (t = −12.04; p < 0.01), and that the net charge is highly correlated with the molecular shape (Figs. 6A and 6B; Fig. S10). Heatmap comparison of the overall shape of calcins and LKTx identified the most evolutionarily labile region of the peptide, noticeable both at sequence and morphological levels: the amino acid residue 29R (in our MSA, 24R in calcin alignment only, Figs. 6C and 6D). This amino acid residue forms the small lateral projection on the bottom of the molecule (absent in the LKTx peptides). Using mutants of maurocalcin, Estève et al. (2003) identified a critical role for this arginine in binding the type 1 RyR (RyR1), because its replacement induced the complete loss of this protein’s effect on RyR.

Figure 5 Correlation test between molecular shape and chemical properties.

Correlation tests between molecular shape and chemical properties (A) Kendall rank correlation between molecular shape (PC both sides), weight, volume and net charge, color of the circle indicates positive or negative correlation coefficient, increasing size of the circle indicates smaller p-value. Kendall rank correlation between Net charge and (B) PC1 front, or (C) PC1 side. r, correlation coefficient.

Figure 6 Evolutionary analyses on the molecular shape and the net charge.

(A) Visualization of net charge values of calcins (purple) and LKTx (orange) as a function of phylogenetic distance (recovered from the dated molecular tree); horizontal axis indicates the time of divergence and vertical axis indicates the net charge values. (B) Phylomorphospace in three dimensions of PC1 values of the frontal and lateral views, and net charge. (C) Heatmap of the frontal (left) and lateral (right) views of calcin and LKTx peptides with major morphological differences represented by warm colors. Outlines represent the mean shape; numbers indicate the amino acid position in the multiple sequence alignment (MSA). (D) MSA of LTKx and calcin peptides, highlighting in squares the amino acid residues responsible for the major morphological differences shown as warm areas on heatmaps (C).

To characterize evolutionary dynamics of ICK peptides, we also tested for significant shifts in morphological regimes using the library l1ou + pBIC (Khabbazian et al., 2016). Eight shifts were detected across the evolutionary history of ICK peptides, of which five were found in the LKTx peptides of parvorder Buthida, and three in the calcins of the parvorder in Iurida (Fig. 7).

Figure 7 Evolutionary shifts in optimum morphology and chemical properties of the toxins.

Shifts in ICK peptide morphology. l1ou and pBIC provided support for eight evolutionary shifts in optimum morphology under an Ornstein–Uhlenbeck (OU) process. Edges with a major morphological evolutionary shift are annotated with a star and bootstrap support (A). Bar graphs showing the seven traits combined: PC1 (B and D) and PC2 (C and E) from the frontal (B and C) and lateral (D and E) shape data; molecular weight (F), molecular volume (G), and net charge (H).

Discussion

Phylogenomic resolution of the scorpion tree of life

Maximum likelihood, species tree analyses, and BI analyses supported the monophyly of the two basal clades in Scorpiones: Buthida and Iurida, a result recovered in a previous, more sparsely sampled phylogenomic analysis (Sharma et al., 2015). The internal relations within Buthidae were congruent with previous morphological hypotheses (Fet et al., 2003; Fet, Soleglad & Lowe, 2005). By contrast, relationships within Iurida were congruent with more recent phylogenomic tree topologies, and maintain the non-monophyly of such groups as Chactoidea (Sharma et al., 2015, 2018). This molecular phylogeny included for the first time a member of the family Hemiscorpiidae, a lineage with a necrotoxic venom (unlike the neurotoxic venom characteristic of most Buthida). Its placement in the scorpion tree of life is nested deeply within a non-monophyletic Scorpionidae, contrary to previous hypotheses based on morphology (Prendini, 2000). This suggests a recent evolutionary origin of necrotoxic venom in scorpions (Fig. 1). The uniformity of these results suggested that the trade-off between missing data, number of genes, and species trees and supermatrix approaches does not have dramatic effects on the reconstruction of the major clades within Scorpiones. The two libraries of Cercophonius squama were not recovered as monophyletic. Given its current taxonomic status and broad distribution (New South Wales, southwestern Australia, Queensland and Tasmania), our results suggest this species could represent a multispecies complex yet to be resolved.

Gene topology of ICK peptides reflects lineage-specific origins of calcins and LKTx

Our survey of ICK homologs greatly expanded their known representation across scorpion phylogeny (Xiao et al., 2016; Santibáñez-López et al., 2016). The gene tree topology recovered herein showed remarkable congruence with the phylogenomic tree (Figs. 1 and 2A). Calcins were recovered as monophyletic and phylogenetically restricted to Iurida. LKTx peptides were not recovered as monophyletic, but were found to be phylogenetically restricted to Buthida (Buthidae, Chaerilidae and Pseudochactidae). Chaerilidae was previously thought to be a member of Iurida, and its placement with Buthidae and Pseudochactidae was previously based only on molecular phylogenetic analyses. Thus, its transfer to Buthida was not substantiated by any morphological characters. The gene tree topology recovered here demonstrates the first known synapomorphy of Buthida as presently defined.

Selection regimes in ICK peptides suggest calcin and LKTx interaction with conserved targets

Some authors (Sollod et al., 2005; Undheim et al., 2015) have suggested that the major problem in reconstructing the molecular evolutionary histories of small peptides, particularly those cysteine-rich peptides, is their conserved disulfide framework, because all non-cysteine amino acids can potentially undergo substitutions without disturbing the protein core. Furthermore, it has been suggested that disulfide bridges, by providing stability, should enable accelerated sequence evolution (by positive selection) and act against deleterious mutations avoiding negative selection (Feyertag & Alvarez-Ponce, 2017). Calcin and LKTx selection analyses showed that non-cysteine amino acids undergo few substitutions in between cysteines, including insertion/deletions (Fig. 2B), and none evolving under positive selection. Thus, our results support the hypothesis that non-CSαβ toxins have evolved under the influence of negative selection to preserve their coding sequences, suggesting they interact with conserved targets (Sunagar et al., 2013). The highly conserved conformation of calcins, along with negative selection to reduce alternate states, is consistent with the tendency for small proteins to rely on being folded into a stable conformation, that is, to preserve their function (venom potency) rather than structural integrity (Undheim, Mobli & King, 2016). This is observed in the fractional conductance induction to RyRs in eight calcins studied (Xiao et al., 2016). The conductance of these calcins ranged from 0.35 to 0.60, suggesting that their high structural similarity allows calcin to bind to RyRs at the same site, engaging the receptive amino acid with varying degrees of affinity.

The high incidence of shifts in Buthida may partly reflect the asymmetrical diversity of this lineage, as the family Buthidae alone comprises approximately half of all described scorpion species. Alternatively, it may reflect some degree of evolutionary lability, as reflected by such metrics as LKTx molecular weight within Buthidae. By comparison, all calcins share a conservative regime (in gray, Fig. 7) except for three non-converging shifts restricted to recently diverging branches. Taken together, our analyses suggest that the evolutionary dynamics of scorpion ICK homologs reflect the underlying phylogeny, not only at the level of sequence data, but also in proxies of protein function.

Calcin evolution within Iurida reflects phylogenetic signal, not classification

According to our phylogenomic analyses, the present classification of Iurida includes numerous non-monophyletic groups (e.g., Chactoidea, Chactidae, Scorpionidae, Hormuridae; Fig. 1). Intriguingly, EFA analysis of calcins revealed a strong correspondence between the principal components of molecular shape and the relationships recovered by our analyses. A vizualization of morphospace within Iurida paralleled the recovery of distinct clades recovered in the phylogenomic tree (Fig. S11). Within the paraphyletic Chactoidea, we observed that each clade that renders the chactoids non-monophyletic clustered in morphospace, frequently to the exclusion of other such clusters, for both the frontal and lateral views. We also examined the morphospace for the relative clustering of Bothriuroidea and Scorpionoidea, which were previously inferred to be part of a monophyletic superfamily (the traditionally defined Scorpionoidea; Prendini, 2000). Consistent with phylogenomic results (Sharma et al., 2015, 2018), Bothriuroidea and Scorpionoidea occupied markedly different parts of morphospace (Fig. S11).

The backbone phylogeny of Iurida was previously not strongly supported and two different phylogenomic datasets recovered different relationships within this group (Sharma et al., 2015; Starrett et al., 2017). Both the paraphyly of Chactoidea and the subdivision of the erstwhile Scorpionoidea have been treated with a reasonable degree of skepticism, given limitations in taxonomic sampling in phylogenomic studies (Sharma et al., 2015, 2018; Starrett et al., 2017; Monod et al., 2017), and thus the classification of scorpions likely retains many non-monophyletic taxa. Our results suggest that calcin sequence, as well as calcin shape, retain high phylogenetic signal in spite of selection (Fig. 2; Fig. S11), and accord with the molecular phylogeny, rather than scorpion classification.

Conclusion

The dataset we assembled here shows that a key single-copy ortholog of an ICK peptide was present in the common ancestor of scorpions. This ancestral peptide subsequently diversified into LKTx and calcins, in the two lineages originating from the basal split in scorpions. Taken together, our analyses suggest that the evolutionary dynamics of the ensuing scorpion ICK homologs reflect to a surprising degree the underlying phylogeny of this arachnid group, not only at the level of sequence data, but also in models of molecular shape and proxies of protein function (e.g., net charge). The evolutionary dynamics exhibited by calcins differ markedly from patterns described in various venomous animals. Comparable analyses of rattlesnake venom have shown that closely related species of Crotalus can possess different subsets of venom genes entirely, a dynamic partly driven by ancient radiation followed by gene loss (Fry et al., 2003; Dowell et al., 2016). Analyses of cone snail and spider venoms have also shown evidence for strong positive selection and extensive gene turnover, with high variance in copy number and elevated rates of non-synonymous mutations (Binford et al., 2008; Garb & Hayashi, 2013; Phuong, Mahardika & Alfaro, 2016; Phuong & Mahardika, 2018). Scorpion ICK homologs exhibit strikingly different phenomena, with little turnover within calcins or LKTxs, and a general pattern of negative selection. The evolutionary conservation of this gene family is all the more remarkable given the estimated Permian age of diversification of crown group scorpions (Sharma et al., 2018).

Our analysis of molecular shape provides the first synapomorphies for the well-supported, and recently redefined, clades Buthida (i.e., the presence of LKTx) and Iurida (i.e., the presence of calcins). These clades have heretofore proven difficult to define using anatomical characters. We were able to show that “molecular morphology” can overcome this limitation. To our knowledge, this work constitutes the first report of a synapomorphy defined from a molecule’s shape in the study of arthropod phylogenetics.

Supplemental Information

Supplemental Information 1 Supplementary Tables.

Click here for additional data file.

Supplemental Information 2 Supplementary Figures.

Click here for additional data file.

We are greatly indebted to Ernesto Ortiz who kindly provided us with information on some of the webservers used here. Access to computing nodes provided by the Center for High Throughput Computing (CHTC) and the Bioinformatics Resource Center (BRC) of the University of Wisconsin-Madison. A subset of the sequencing was performed by Caitlin M. Baker and Julia Cosgrove (Giribet Lab, Harvard University). Comments from Kevin Arbuckle and two anonymous reviewers refined an earlier draft of the manuscript.

Additional Information and Declarations

Competing Interests

Author Contributions

DNA Deposition

Data Availability

The authors declare that they have no competing interests.

Carlos E. Santibáñez-López conceived and designed the experiments, performed the experiments, analyzed the data, prepared figures and/or tables, authored or reviewed drafts of the paper, approved the final draft.

Ricardo Kriebel performed the experiments, analyzed the data, prepared figures and/or tables, authored or reviewed drafts of the paper, approved the final draft.

Jesús A. Ballesteros performed the experiments, analyzed the data, prepared figures and/or tables, authored or reviewed drafts of the paper, approved the final draft.

Nathaniel Rush performed the experiments, approved the final draft, elaborated, updated and maintained the scorpion database.

Zachary Witter performed the experiments, approved the final draft, elaborated, updated and maintained the scorpion database.

John Williams performed the experiments, approved the final draft, elaborated, updated and maintained the scorpion database.

Daniel A. Janies performed the experiments, contributed reagents/materials/analysis tools, authored or reviewed drafts of the paper, approved the final draft, elaborated, updated and maintained the scorpion database.

Prashant P. Sharma conceived and designed the experiments, performed the experiments, analyzed the data, contributed reagents/materials/analysis tools, authored or reviewed drafts of the paper, approved the final draft.

The following information was supplied regarding the deposition of DNA sequences:

All sequences are deposited in GenBank, accession numbers: Urodacus elongatus, SRR7885472; Kolotl magnus, SRR7879236.

The following information was supplied regarding data availability:

Santibanez, Carlos; Kriebel, Ricardo; Ballesteros, Jesus; Rush, Nathaniel; Witter, Zachary; Williams, John; et al. (2018): calcin_data. figshare. Fileset. https://doi.org/10.6084/m9.figshare.5686633.v1

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
