# Peer review of "Integration of phylogenomics and molecular modeling reveals lineage-specific diversification of toxins in scorpions"

_PeerJ, doi:10.7717/peerj.5902_

## Round 0.1 · original submission · Minor Revisions

I have now received three excellent reviews of your paper and they are all very enthusiastic recommending minor revisions. They have all detailed a number of little issues that you'll need to deal with to clean up the paper, but then you should be good to go. Congratulations.

Reviewer 1 ·

Basic reporting

I enjoyed reading this manuscript. You did a good job of clearly outlining your goals, thoroughly explaining the methods and results, and putting the findings within a relevant broader context. I was particularly impressed with the extensive and rigorous methods. I think it is a creative approach for using genomic data to better understand the phylogenetic patterns in a group of organisms. The figures and tables are generally informative and look nice. The raw data is easily accessible. Nevertheless, I do have some suggestion that I think will improve the manuscript.

- It would have been helpful to see more general background information on scorpion taxonomy in the introduction. How many species are there? You do a good job of referencing past phylogenetic work in scorpions, but there isn’t much context for statements about taxon sampling. Are 55 species a good representation?

- In a few places in the introduction it is unclear if you are discussing toxins/venoms more generally or specific to scorpions. Line 55 is a good example of this. I assumed you were talking about scorpion venoms, but it’s not completely clear.

- Figure 1: I think you should add divergence estimates on the tree. You make mention of some dates in the text, and it would be nice to reference these estimates and others across the group. Also, it’s not clear to me what the bars to the right of the tree indicate. Are they the number of genes in each taxon? Forgive me if I simply missed something in the caption.

- I don’t know if Figure 2 is necessary. It doesn’t seem to be adding any information that isn’t already in Figure 1. Perhaps it could be moved to the supplement?

- The caption to Figure 4 doesn’t have an explanation of panels c and d.

- In Lines 104-106, could you give a breakdown of the data you used? How many were your transcriptomes, complete scorpion genomes, and venom libraries? It’s in TableS1, but it would be nice to have just a few numbers here in the methods.

- The manuscript is generally well written, but there are several typos and confusing/awkward phrases. I’ve made note of them in the additional comments below.

Experimental design

The experimental design, particularly the phylogenomic methods, seem solid to me. I have a few relatively minor comments:

- Lines 119-120, 173: What were your parameters for MAFFT and trimAl?

- Lines 142-143: I could not find the Bayesian dating results, even though you stated these were explored in this study. Perhaps you could add a dated Bayesian tree to the supplement?

- Lines 160-161: Why did you only run PhyloBayes on Matrix 2? Computational time? I think it would be best to exclude this analysis from the paper altogether. It doesn’t add much other than to say “we did it.”

Validity of the findings

I thought the data and approach in this manuscript were very sound. The findings seem to be an important contribution to scorpion systematics and venom evolution more generally.

I do think the conclusion section can be expanded. As currently written, the conclusions focus only on molecular morphology as a synapomorphy for these particular groups of scorpions. However, this doesn’t exactly connect well with the abstract/introduction, which sets up the paper as addressing venom evolution. Basically, most of the paper seems to be moving towards using the scorpion phylogeny to understand the evolution of scorpion venom (see the first sentence of the abstract), but then concludes that you used scorpion venom to help resolve the scorpion phylogeny. I think you should add some conclusions and/or more discussion about how this phylogenetic hypothesis is important for our understanding of venom evolution. This would tie things nicely back to the intro.

I also have another comment regarding Figure 1. If the bars to the right of the tree are indeed the number of genes from each taxon (see above question), are you concerned that some taxa have so few genes? Some of less than 10. It’s also curious that so many taxa in Buthida are missing lots of genes. Any reason for this, and if so, perhaps devote space to this in the results and discussion?

Additional comments

Here are some additional minor comments, mostly related to typos and unclear sentences:

Line 24: say “phylogenomic relationships of scorpions” instead.

Line 39: Perhaps avoid using the word “fascinating” in consecutive sentences.

Lines 46-47 is worded awkwardly. I think the phrase “recruitment from housekeeping gene into venom” is confusing.

Line 69: say “contains” rather than “bears.”

Lines 80-81 and 84: I don’t think you need to say “taxonomic samping to…” Just say “limited to 25 species” and “limited to six species.”

Line 83. The term “non-overlap(ping) here and in the previous sentence is a little confusing. I understand what you’re saying, but I had to read it twice. Something like “focused on different species” and “used a different molecular dataset” would clarify things.

Line 105: “published” or “existing complete scorpion genomes…”

Line 121: “less than 50 amino acids…”

Line 147: What is “Matrix 1?” Maybe just say “our original matrix (Matrix 1).”

Line 150: “using a different number of species…”

Line 166: “well as from GenBank and…”

Line 278: “were represented by 52…”

Line 289-290: The sentence is a little awkward. Perhaps change to something like, “To detect the direction of selection acting on the codon…”

Line 291: No need for the “While” at the beginning of the sentence.

Line 365: should be “clades.”

Figure 6 caption: indicates greater or smaller p-value?

Figure 7 caption: “amino acid position in the multiple sequence alignment.”

Reviewer 2 ·

Basic reporting

no comment

Experimental design

no comment

Validity of the findings

no comment

Additional comments

This manuscript is pretty solid and straightforward. The analyses are well-done and the manuscript as a whole is clear. While I am not an expert in toxins, the choice of analyses seems justified, are not difficult to understand or interpret, and the results are clear. The criteria for Basic Reporting, Experimental Design, and Validity of Findings are all met. The figures are clear and quite eye-catching. I feel this manuscript is acceptable for publication, given some minor edits, which I include below.


Minor comments and recommendations:

Line 104-105: “generated by us”. This is misleading since it seems that the vast majority of transcriptomes used were previously published by others. It seems only two transcriptomes were newly generated for this study. This should be clarified.

Line 147: “maximizing” to “maximize”

Line 150 and elsewhere: although “orthologues” is acceptable spelling, “orthologs” is more commonly used.

Line 265: “with previous hypothesis”. Which one? And it should be “with a previous hypothesis” or “with previous hypotheses”.

Line 272: I’ve never personally used phylobayes, but I hear about analyses failing to reach convergence with this program quite a bit. Honestly, I don’t think this analysis is even needed. Your other analyses are pretty well supported and congruent. It’s great that you want to report it, but it wouldn’t bother me to just cut out mentions of phylobayes.

Line 278: “were represented by 52”

Line 312: “Morphometric analysis of molecular”. Should this refer to Figure S3?

Figure 1. What are the numbers next to the phylogeny?

Figure 4. the axis labels could be bigger.

·

Basic reporting

This is generally of a high standard, there are mostly just a few typos scattered across the text (listed below). Make sure that you present statistical results in full, it is not sufficient to just present P-values as is done throughout the results. Otherwise I would recommend the authors include their species tree on Figshare as well as the other resources available there. The authors have included the data needed to reconstruct this tree but it is fairly standard in the field to provide the tree as well as this enables readers to check details and replicate particular results without requiring the time-consuming step of building the tree again. Finally, the references are inconsistently formatted throughout in terms of issues such as capitalisation of article and journal titles, and the presence of a DOI. This should be standardised throughout. Part of the Coddington et al. reference seems to be missing and with the detail given I'm not sure what the Dongen reference is. There are also some issues with reference order, e.g. Santibanez-Lopez et al. (2017) is listed before Santibanez-Lopez et al. (2016), and Sharma et al. 2015 is listed before Sharma et al., 2016.

TYPOS AND SIMILAR MINOR ERRORS
ln 81 and 90 - Be careful about the use of 'significant' when not referring to statistical results as this can create confusion for readers.

ln 121 - 'less 50' should be 'less than 50'

ln 147 - maximizing should be maximize

ln 225-226 - the correct reference for phytools is Revell, L.J., 2012. phytools: an R package for phylogenetic comparative biology (and other things). Methods in Ecology and Evolution, 3(2), pp.217-223

ln 245 - http://venom.space doesn't open for me, it just takes me to a login box with no option to create an account

ln 327 - 'conservative compared to what? The figures refered to here are open to subjective interpretation in support of this statement - for instance to me it looks like a reasonable amount of evolution has happened, so make sure you qualify this.

ln 346 - Fig. 6 doesn't need the reference to parts b and c

ln 350 - Reference to figure 7 has parts c and listed out of alphabetical order for some reason.

ln 365 - 'caldes' should be 'clades'

ln 375-378 - Nice point, but where your two libraries derived from different populations (if so where from)?

ln 395 - I think 'cysteine-rich like peptides' should just be 'cysteine-rich peptides'

ln 440 - 'retain' should be 'retains'

Figure 1 - what do the bars (and numbers) beside the phylogeny represent?

Figure 2 - 'Sensitive' should be 'sensitivity'. I'm also not sure from the descriptions and keys how the Navajo plots in Figures 1 and 2 actually differ other than in colour scheme - there are more plots in Figure 2 but those that overlap with the ones in Figure 1 seem identical (further adding to the confusion).

Figure 3 - Should the last sentence be structured with plurals (i.e. amino acids, their)?

Figure 4 - The legend refers only to parts a and b but the figures are labelled with a-d (I think a and b correspond to a in the legend and c and d correspond to b).

Figure 6 - 'color of the circle indicates greater correlation coefficient' could lose the word 'greater' and improve clarity, and 'size of the circle indicates greater P-value' should, I think, be 'increasing size of the circle indicates smaller P-values'

Figure 7 - 'date' should be 'dated', 'axe' should be 'axis'

Experimental design

Very well done throughout, in particular it is clear that careful attention has been paid to justifying analytical decisions.

The one exception is that at two points in the methods (ln 106 and 139-141) the authors refer to previous papers for details. Although this happens semi-regularly in the literature, I believe that readers should be able to understand what has been done in a paper without having to refer to several other papers to figure it out. Please give at least brief details here, and preferably a full account of methods.

I would also point out that the authors refer to "Matrix1" in ln 147 but don't actually explain what this is until the results section (ln 259-260).

Validity of the findings

Both data and results appear of high quality and the conclusions do not stray from what is justifiable from the results themselves. I might only suggest an idea to the authors, which they are free to ignore or incorporate depending on whether they think it has enough merit to be worth discussing. Your results seem to be consistent with the ancestral scorpion possessing an ICK toxin that is ancestral to both LKTx and calcins (which subsequently diversified into these groups in the two lineages originating from the basal split in scorpions). To me that would be a very interesting suggestion and might be one you want to make in the manuscript.

Additional comments

I would finally just like to congratulate the authors on an excellent piece of work that I greatly enjoyed reading. With only the minor issues above address, this paper is definitely worthy of publication.

Best wishes,
Kevin Arbuckle

---

## Round 0.2 · accepted · Accept

Thank you for your careful revision that, in my view, addresses nicely the previous concerns of your manuscript. I am now happy to recommend acceptance of your paper. However, I do want to point out that despite twice in your rebuttal letter that you claim your data are now accessible at the indicated website (venom.space), this website is STILL inaccessible. This needs to be corrected before your paper can move forward. Additionally, I would request that you submit your transcriptome data to GenBank (or some other public repository) and provide appropriate accession numbers. All too often, lab website go by the wayside very quickly and data become inaccessible as a consequence when only hosted locally.

#